# Circular olefin copolymers made de novo from ethylene and α-olefins

Xing-Wang Han [1], Xun Zhang[2], Youyun Zhou[1], Aizezi Maimaitiming[2], Xiu-Li Sun [2], Yanshan Gao [2] ✉, Peizhi Li[1], Boyu Zhu[1], Eugene Y.-X. Chen [3] ✉, Xiaokang Kuang[1] & Yong Tang [1,2] ✉

Ethylene/α-olefin copolymers are produced in huge scale and widely used, but their after-use disposal has caused plastic pollution problems. Their chemical inertness made chemical re/upcycling difficult. Ideally, PE materials should be made de novo to have a circular closed-loop lifecycle. However, synthesis of circular ethylene/α-olefin copolymers, including high-volume, linear low-density PE as well as high-value olefin elastomers and block copolymers, presents a particular challenge due to difficulties in introducing branches while simultaneously installing chemical recyclability and directly using industrial ethylene and α-olefin feedstocks. Here we show that coupling of industrial coordination copolymerization of ethylene and α-olefins with a designed functionalized chain-transfer agent, followed by modular assembly of the resulting AB telechelic polyolefin building blocks by polycondensation, affords a series of ester-linked PE-based copolymers. These new materials not only retain thermomechanical properties of PE-based materials but also exhibit full chemical circularity via simple transesterification and markedly enhanced adhesion to polar surfaces.

Polyethylene (PE)-based materials, including ethylene homopolymer and its random or block copolymers with α-olefins, are the most produced synthetic materials globally, but current practices in their production, use, and after-use that follow the linear materials economy framework have taken a huge toll on both the environment and society[1,2]. Although several notable advances have been made in PE chemical re/upcycling[3–10], energy-efficient and selective catalytic processes are still lacking due to the inherent chemical inertness of C-C and C-H bonds in PE[11]. Ideally, PE materials should be made de novo to have a circular, closed-loop lifecycle[12–16]. Recently, several circular polymers with closed-loop lifecycles were developed[17–23], including recyclable high-density PE (HDPE)-like polymers[24–26]. Several notable recent advances were made towards circular HDPE-like polymers with closed-loop chemical recyclability through incorporation of cleavable linkages such as ester bonds into the PE backbones. In 2021, long-chain alkyl polyester produced via polycondensation of diol or diester

monomers[27,28] was reported to possess HDPE-like thermomechanical properties[23]. In 2022, another approach to circular HDPE-like polyester was developed using the diester end-capped PE obtained via tandem dehydrogenation/metathesis of post-consumer HDPE[25] or tandem ethylene copolymerization with an oxa-norbornene comonomer, retro-Diels-Alder reaction, and metathesis[26] (Fig. 1b). As compared to HDPE, ethylene/α-olefin copolymers produced by direct copolymerization of ethylene and α-olefins have a broader range of applications as well as high-value polyolefin elastomer (POE) and robust olefin block copolymer (OBC) thermoplastic elastomer (plastomer) materials, but chemical or mechanical recycling of such copolymers is considerably more challenging due to their chemical and structural heterogeneity. Hence, there is a pressing need to render such materials chemically circular but achieving such a desirable goal presents a particular challenge, due to difficulties in introducing controllable branches, while simultaneously installing chemical circularity, and also in

[1]Shenzhen Grubbs Institute, Southern University of Science and Technology, Shenzhen 518055, China. [2]State Key Laboratory of Organometallic Chemistry, Shanghai Institute of Organic Chemistry, Chinese Academy of Sciences, Shanghai 200032, China. [3]Department of Chemistry, Colorado State University, Fort Collins, CO 80523-1872, USA. ✉e-mail: gaoyanshan@sioc.ac.cn; eugene.chen@colostate.edu; tangy@sioc.ac.cn

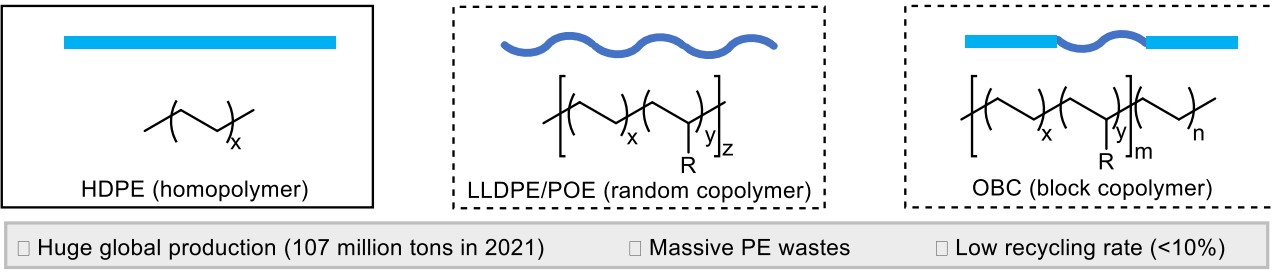

**a  Various commercial PE materials**

HDPE (homopolymer)

LLDPE/POE (random copolymer)

OBC (block copolymer)

☐ Huge global production (107 million tons in 2021)    ☐ Massive PE wastes    ☐ Low recycling rate (<10%)

**b  Recent progress on circular HDPE-like polymers (*23-25*)**

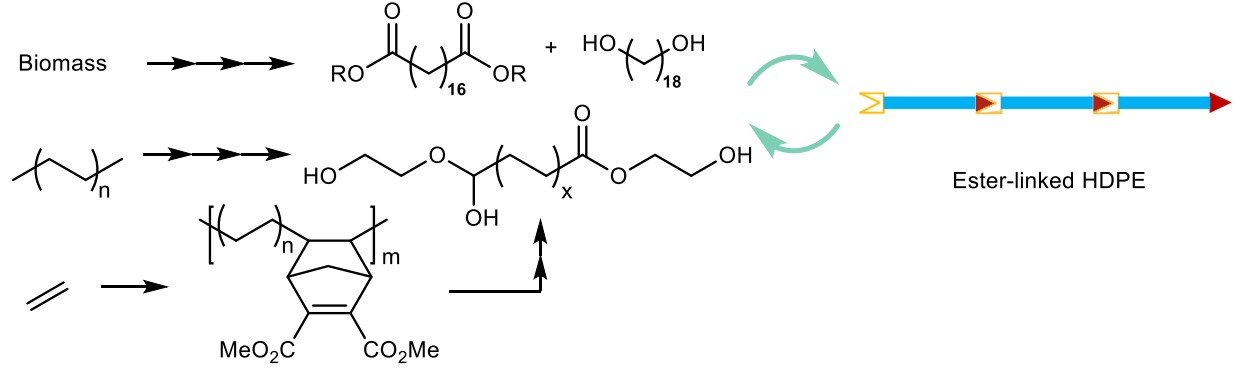

Biomass

Ester-linked HDPE

**c  This work: Circular olefin copolymers**

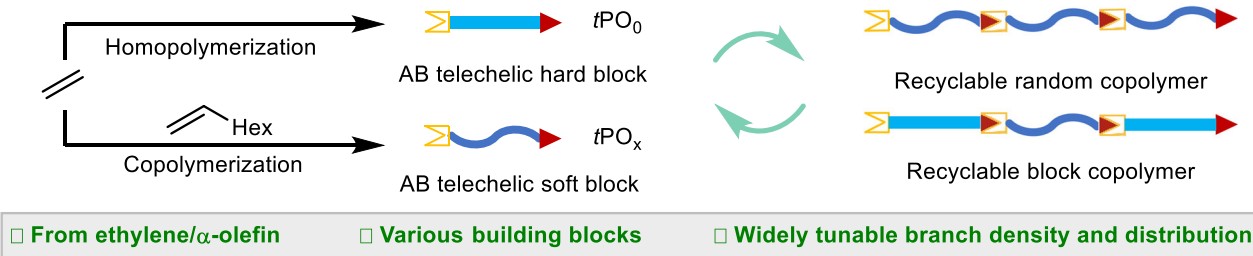

Homopolymerization

$tPO_0$

AB telechelic hard block

Hex

Copolymerization

$tPO_x$

AB telechelic soft block

Recyclable random copolymer

Recyclable block copolymer

☐ **From ethylene/α-olefin**    ☐ **Various building blocks**    ☐ **Widely tunable branch density and distribution**

**Fig. 1 | Conventional PE materials and circular alternatives. a** Various commercial PE materials such as HDPE, LLDPE, POE and OBC, and associated plastics problems. **b** Selected examples of HDPE-like polyesters with closed-loop recycling and their routes. **c** This work: circular ethylene/α-olefin copolymers, including LLDPE, POE and OBC materials, produced de novo from ethylene and 1-octene via a combined CCTP and polycondensation method. Note that the blocks represent macromolecular chains with a distribution of length.

primarily using industrially abundant and inexpensive feedstocks, ethylene and α-olefins such as 1-hexene or 1-octene.

We reasoned that an effective and scalable approach should be an innovative drop-in process based on the widely practiced coordination (co)polymerization of ethylene and α-olefins in the current plastics industry[29]. To realize this strategy, we first synthesize AB-type telechelic PE building blocks end-capped with alcohol (OH) functionality (A) at one end and ester (CO$_2$Et) functionality (B) at the other by coordinative chain-transfer polymerization (CCTP) of ethylene and 1-octene with suitable chain-transfer and chain-end capping reagents[30]. We further hypothesized that the advantage of AB telechelic building blocks over A2 + B2 ones in polycondensation, especially concerning precise stoichiometric control, should allow for Lego-type modular assembly of AB building blocks into on-demand PE materials. Accordingly, polycondensation of these AB-type telechelic PE building blocks of various molecular weights and 1-octene incorporations leads to circular LLDPE, POE, and OBC materials on-demand, with their closed-loop recycling efficiently achieved in quantitative interconversions between telechelic building blocks and high-molecular-weight ester-linked polyolefins by repeated polycondensation/

depolymerization processes (Fig. 1c). Compared with traditional LLDPE, POE and OBC materials, these circular alternatives not only bring in closed-loop recyclability, while without compromising thermomechanical performance, but also exhibit significantly enhanced adhesion properties to polar surfaces.

## Results
### Design and synthesis of AB telechelic polyolefin building blocks (*t*PO)

The challenge in synthesizing circular polymers using ethylene and an α-olefin as the primary feedstocks is to develop a catalytic process that can produce AB-type telechelic macromonomers with tailored molecular weight and high purity, which serve as the highly modular building blocks for synthesizing high-molecular-weight polymers. To obtain such telechelic macromonomers, complementary functional groups such as alcohol and ester suitable for subsequent assembly should be introduced at each chain end (i.e., AB telechelic). An alkyl metal reagent (R$_n$M) which contains a functional group (FG') at the end of alkyl chains is typically used as a functionalized chain transfer agent (*f*CTA) in the transition-metal-catalyzed CCTP to synthesize telechelic polymers[31,32].

Here we designed and synthesized an alkyl zinc reagent $Zn[(CH_2)_6OTIPS]_2$ (TIPS = triisopropylsilyl) as the $f$CTA for the ethylene/1-octene CCTP. Each time the $Zn$-$(CH_2)_6OTIPS$ exchanges with a catalytically active Zr-R (R = Bn, polymeryl) species, the resulting Zr-$(CH_2)_6OTIPS$ initiates a polymer chain (Zr-polymeryl, polymeryl indicates a polymer chain with an OTIPS group at the chain head), and the $Zn$-$(CH_2)_6OTIPS$ group can exchange with Zr-polymeryl during the polymerization process. The Zn-polymeryl can continue to undergo chain transfer as confirmed by the continuously growing polymer $M_w$ over reaction time (entries 1–2, Table 1). The chain transfer efficiency as gauged by the number of polymer chains generated by each $R_2Zn$ (chains/Zn) is between 1.3 and 1.8. When the polymerization is complete, the obtained $Zn(PO)_2$ reacts with a quenching reagent, ethyl succinyl chloride ($ClCOCH_2CH_2CO_2Et$), to introduce -$COCH_2CH_2CO_2Et$ to the other chain end (Table 1). After the silyl group deprotection with HCl/EtOH, a series of AB telechelic PO macromonomers with varied molecular weight and branch density were obtained, through tuning the CCTP reaction by adjusting the relative concentration of 1-octene comonomer and the reaction time (Table 1). For example, entry 1 shows that 1.8 g $t$PO was obtained via a typical ethylene homopolymerization with 20 $\mu$mol catalyst, 22 $\mu$mol cocatalyst, and 1.0 mmol $Zn[(CH_2)_6OTIPS]_2$ (50 equiv./cat) as the $f$CTA in toluene for 3 min under 5 atm ethylene pressure. After silyl deprotection, $t$PO$_0$–1 was obtained with $M_w$ = 0.8 kg/mol and $M_n$ = 0.6 kg/mol, which is close to the value calculated based on $^1$H NMR ($M_n$ = 0.9 kg/mol, Supplementary Fig. 4). The ratio of the polymer chains with the functional groups at both chain ends (quenching efficiency) is about 95% as determined by $^1$H NMR. A series of AB telechelic, -OH and -COOEt end-capped ethylene/1-octene copolymers with widely tunable molecular weight and branch density values were synthesized (Table 1, entries 2–7). The telechelic macromonomers ($t$PO) with $M_n$ ranging from 1.6 to 6.1 kg/mol and $Đ$ in a narrow range of 1.2 to 1.4) were obtained by varying the reaction time from 5 to 30 min. The 1-octene incorporation can also be efficiently tuned from 3.1% to 14.8% by changing the 1-octene comonomer concentration (Table 1), showing the desired tunability of the telechelic PO building blocks.

Note that the synthesis of telechelic polyolefin with high difunctional purity based on CCTP is challenging. The catalyst activation could introduce Bn- or H- chain head for the chain initiation; the undesired $\beta$-H transfer could introduce olefinic chain end; while quenching reaction with acyl chloride is very efficient and selective, it could possibly still introduce proton (moisture or HCl) and lead to -CH$_3$ chain end. The -CH$_3$ chain end is the major type of the impurity source. While the quenching efficiency is high in ethylene homopolymerization, the ethylene/1-octene copolymerization affords telechelic macromonomers with reduced purity. In the latter case, a flash chromatography was conducted for purification of the telechelic copolymers, which efficiently removed the impurities and achieved high difunctional purity. Furthermore, the low molecular weight telechelic macromonomers $t$PO$_0$–1 (entry 1), $t$PO$_0$–2 (entry 2) and $t$PO$_{14.8}$ (entry 6) were chosen as representative samples for MALDI-TOF analysis to confirm the microstructure and chain end functionalities (Supplementary Figs. 50 to 52). The results unambiguously suggested the successful synthesis of the telechelic macromonomers. Note that cyclic dimer of $t$PO$_0$–2 was detected, and we failed to observe the methyl-terminated macromonomer as the minor impurity, which likely reflects the MALDI-TOF bias on evidencing chain-end fidelity especially for the samples with high purity.

### Synthesis of recyclable olefin random copolymer ($r$PO) and block copolymer ($r$OBC)

With these AB telechelic building blocks in hand, we conducted polycondensation reactions to synthesize ester-linked, ethylene-based random polyolefin copolymer $r$PO and block copolymer $r$OBC. This Lego-inspired modular assembling method provides more ideal tunability in the synthesis of ethylene copolymers with microstructures analogous to commercial PE with a few degradable ester linkages along the main chain.

The polycondensation was typically conducted at 190 °C under vacuum for 24 h. The AB building blocks can be assembled into higher molecular weight $r$PO with $M_w$ ranging from 85.7 to 239.8 kg/mol with $Đ$ ranging from 2.3 to 5.5 (Table 2). The polycondensation reactions were efficient, affording near quantitative isolated yields. The $M_w$ of $r$PO$_0$–1 can reach up to 122.3 kg/mol with $Đ$ = 3.4 (entry 1, Table 2). Using $t$PO$_0$–2 and $t$PO$_{3.1}$ as the building blocks, $r$PO$_0$–2 ($M_w$ = 85.7 kg/mol, $Đ$ = 3.2, entry 2) and $r$PO$_{3.1}$ ($M_w$ = 113 kg/mol, $Đ$ = 3.6, entry 3) were synthesized. Using AB telechelic building blocks with different 1-octene contents, $t$PO$_x$ (x = 8.9 – 14.8) for polycondensation, $r$PO$_x$ copolymers with $M_w$ up to 240 kg/mol were obtained (entries 4 to 7). Note that minimal Ti residue was detected by ICP-MS in the obtained polymer samples, <0.001 wt% for two representative samples of entries 1 and 7 in Table 2, suggesting the efficient removal of Ti metal residue after the post-polycondensation procedure.

The development of efficient catalytic methods for synthesizing OBC plastomers with hard and soft blocks has been of great interest to both industry and academia[33,34]. The synthesis of ethylene/α-olefin OBCs through coordination polymerization is typically limited to the chain shuttling polymerization[35–37] or the stepwise/sequential reaction methods[38,39]. In contrast to these chain-growth olefin polymerization methods, our AB telechelic building block strategy offers unique flexibility for the modular synthesis of ester-linked block copolymers with a combination of hard and soft telechelic building blocks via polycondensation. For example, $r$OBC$_{7.7}$ with high $M_w$ (129.4 kg/mol, $Đ$ = 2.8) and 7.7 mol% 1-octene was obtained by the reaction of $t$PO$_{3.1}$ (hard block) and $t$PO$_{9.6}$ (soft block) in 0.6:2.4 weight ratio under polycondensation conditions (entry 8, Table 2). Switching the building blocks to $t$PO$_0$–1 and $t$PO$_{14.8}$ with a weight ratio of 0.6:1.1, $r$OBC$_{9.4}$ ($M_w$ = 99.4 kg/mol) was obtained with a branch density/octene content of 9.4 mol% (entry 9, Table 2).

### Properties of ester-linked olefin copolymers $r$PO and $r$OBC

Previous studies have shown that the long-spaced aliphatic polyesters show similar thermal and mechanical properties to HDPE, and the properties become more similar with longer CH$_2$ spacings[28,40,41]. Here we examined thermal and mechanical properties as well as adhesion to polar surfaces of the ester-linked ethylene random and block copolymers $r$PO and $r$OBC, compared to the properties of related conventional ethylene-based polyolefin ($c$PO) materials.

**Thermal Properties.** The $r$PO and $r$OBC samples with various microstructures exhibited widely tunable thermal properties with melting behavior, low temperature resistance, and thermal stability mimicking those of the commercial PE. The melting temperature ($T_m$) of the HDPE analog $r$PO$_0$–1 and $r$PO$_0$–2 is 115.2 °C and 124.4 °C, respectively, as measured by differential scanning calorimetry (DSC; Fig. 2a). The $T_m$ of $r$PO$_{3.1}$ is 106.9 °C, which is typical for commercial LLDPE. The $T_m$ of the $r$PO copolymer decreases from 87.5 °C to 51.0 °C with increasing the branch density/octene content ($r$PO$_{8.9}$ to $r$PO$_{14.8}$), which is typical for commercial POE products (e.g., $T_m$ = 74.5 °C for $c$PO$_{11.2}$)[29]. Notably, the $T_m$ of multiblock $r$OBC$_{7.7}$ and $r$OBC$_{9.4}$ is 95.1 °C and 101.1 °C, corresponding to the hard building blocks, $t$PO$_{3.1}$ and $t$PO$_0$–1, respectively. The low glass transition temperature ($T_g$) of $r$OBC$_{7.7}$ ($T_g$ = −44.7 °C) and $r$OBC$_{9.4}$ ($T_g$ = −54.4 °C) also reflects on their characteristic elastic properties of the soft building blocks, $t$PO$_{9.6}$ and $t$PO$_{14.8}$, respectively[35]. Further powder X-ray diffraction (pXRD) studies on these samples showed similar solid-state structure to analogous commercial PE materials, with the crystalline and amorphous phases essentially retained in all the samples. The same diffraction peaks and patterns typical for HDPE and LLDPE were also observed in $r$PO$_0$–1, $r$PO$_0$–2, and

**Table 1 | The synthesis of AB-type telechelic building blocks via ethylene/1-octene CCTP[a]**

| Entry | Octene (mmol) | t (min) | $tPO_x$ | Wt.[b] (g) | Act.[c] | Octene content (mol%)[d] | difunctional purity (%)[e] | $M_n$[f] (NMR) | $M_n$[g] (GPC) | Đ[g] | Chains/Zn (NMR) |
|---|---|---|---|---|---|---|---|---|---|---|---|
| 1 | - | 3 | $tPO_0\text{-}1$ | 1.8 | 360 | 0 | 98 | 0.9 | 0.6 | 1.4 | 1.6 |
| 2 | - | 8 | $tPO_0\text{-}2$ | 4.6 | 345 | 0 | 91 | 3.1 | 2.1 | 1.4 | 1.8 |
| 3 | 10 | 15 | $tPO_{3.1}$ | 8.4 | 336 | 3.1 | 96 | 4.4 | 3.3 | 1.2 | 1.8 |
| 4 | 40 | 20 | $tPO_{9.6}$ | 10.2 | 306 | 9.6 | 99 | 7.5 | 4.8 | 1.3 | 1.4 |
| 5 | 40 | 30 | $tPO_{8.9}$ | 13.2 | 264 | 8.9 | 99 | 8.4 | 6.2 | 1.3 | 1.6 |
| 6 | 60 | 5 | $tPO_{14.8}$ | 3.4 | 408 | 14.8 | 99 | 2.7 | 1.6 | 1.2 | 1.3 |
| 7 | 60 | 20 | $tPO_{12.2}$ | 11.2 | 336 | 12.2 | 99 | 7.1 | 5.1 | 1.4 | 1.6 |

[a]Step 1, CCTP; 20 μmol catalyst Zr[tBu-ONPyrO]Bn$_2$, 22 μmol co-catalyst (C$_{16}$H$_{33}$)$_2$NPhH⁺B(C$_6$F$_5$)$_4$⁻, 1.0 mmol $f$CTA; 5 atm ethylene. Step 2: EtOCOCH$_2$CH$_2$COCl (6.0 equiv.), 120 °C, N$_2$, overnight. Step 3: HCl/EtOH. For clarity, the telechelic polyolefin (PO) macromonomer was defined as $tPO_x$ with $t$ representing *telechelic* and the subscripted × indicating the octene content in mol% in the copolymer.
[b]Isolated yield after precipitation in MeOH.
[c]Activity in kg/(mol cat.)·h·atm.
[d]Determined by ¹H NMR, calculated without taking chain end functional groups into consideration.
[e]Determined by ¹H NMR, the difunctional purity was measured after the purification (except $tPO_0$–2) and silyl deprotection steps.
[f]Determined by ¹H NMR.
[g]Number-average ($M_n$) molecular weights and dispersity (Đ) determined by GPC, kg/mol.

**Table 2 | Polycondensation of AB telechelic building blocks to *rPO* and *rOBC*[a]**

| entry | polymer | AB block(s); ratio | $M_w$ [b] | $M_n$ [b] | $Đ$ [b] | octene content (mol%) [c] | ester groups/chain [d] | $T_m$ (°C) [e] | $T_g$ (°C) [e] | Crystallinity [e] |
|---|---|---|---|---|---|---|---|---|---|---|
| 1 | $rPO_0$-1 | $tPO_0$-1 | 122.3 | 36.0 | 3.4 | 0 | 60.0 | 115.2 | n.d. | 46.7 |
| 2 | $rPO_0$-2 | $tPO_0$-2 | 85.7 | 26.8 | 3.2 | 0 | 12.8 | 124.4 | n.d. | 56.3 |
| 3 | $rPO_{3.1}$ | $tPO_{3.1}$ | 113.4 | 31.5 | 3.6 | 3.1 | 8.8 | 106.9 | n.d. | 39.8 |
| 4 | $rPO_{8.9}$ | $tPO_{8.9}$ | 100.9 | 40.4 | 2.5 | 8.9 | 6.6 | 87.5 | −44.1 | 25.9 |
| 5 | $rPO_{9.6}$ | $tPO_{9.6}$ | 90.6 | 39.4 | 2.3 | 9.6 | 7.9 | 77.0 | −48.5 | 15.9 |
| 6 | $rPO_{12.2}$ | $tPO_{12.2}$ | 110.1 | 44.0 | 2.5 | 12.2 | 8.6 | 51.0 | −51.6 | 10.6 |
| 7 | $rPO_{14.8}$ | $tPO_{14.8}$ | 239.8 | 43.6 | 5.5 | 14.8 | 27.3 | 54.5 | −51.1 | 8.2 |
| 8 | $rOBC_{7.7}$ | $tPO_{3.1}/tPO_{9.6}$ 0.6/2.4 | 129.4 | 46.2 | 2.8 | 7.7 | 10.4 | 95.1 | −44.7 | 24.3 |
| 9 | $rOBC_{9.4}$ | $tPO_0$-1/$tPO_{14.8}$ 0.6/1.1 | 99.4 | 32.1 | 3.1 | 9.4 | 31.8 | 101.1 | −54.4 | 20.5 |

[a]Conditions: The telechelic macromonomer in a flask was heated at 60 °C under vacuum for 30 min. A toluene solution of Ti(O$^n$Bu)$_4$ (0.05 mol%) was added, then the temperature was raised to 190 °C. Vacuum was gradually applied (600 mbar to 2 mbar) over 3 h. Typically, the polymerization was conducted at 190 °C under vacuum for 24 h (see Supplementary Information for more details).
[b]Determined by GPC, kg/mol.
[c]Determined by $^1$H NMR spectroscopy.
[d]Number of ester groups per chain (ester/chain) = $M_n(rPO)/M_n(tPO)$ for *rPO* (entries 1–7); $M_n(rPO)/M_n(tPO)$ for *rOBC* (entries 8–9): $\bar{M}_n$ (tPO) = [$M_n(tPO_A)×f_A+M_n(tPO_B)×f_B$] ÷($f_A+f_B$); $f_A$ and $f_B$ indicate the mole fraction of the respective telechelic macromonomer.
[e]Determined by DSC, second heating cycle.

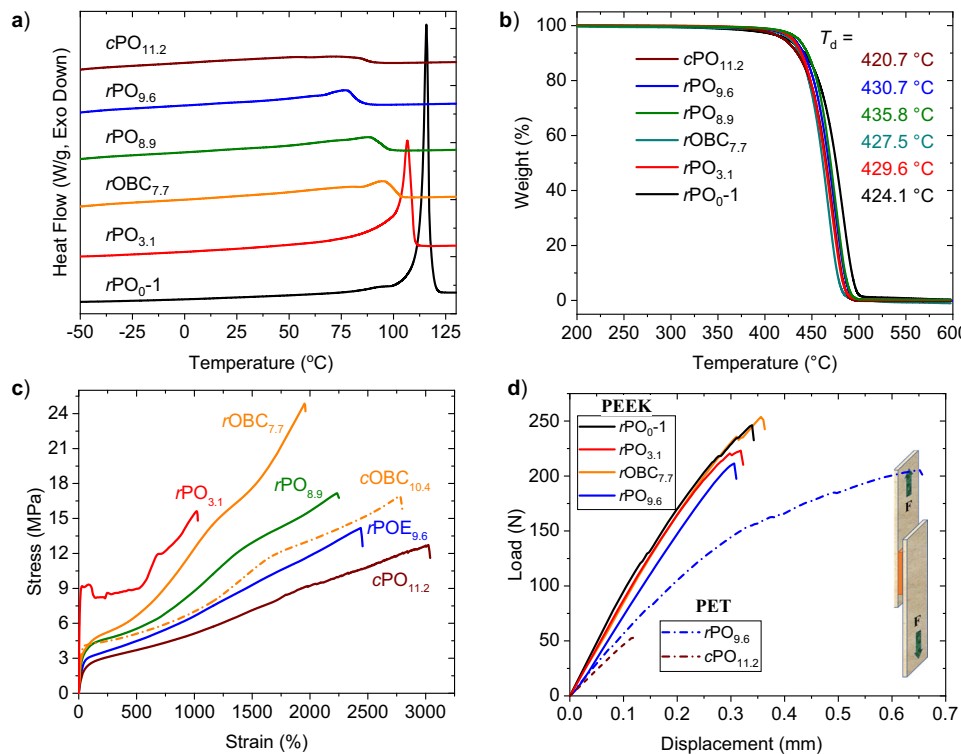

**Fig. 2 | Thermal and mechanical properties of *r*PO and *r*OBC materials. a** Melting curves of *r*PO and *r*OBC samples by DSC, second heating scan. **b** TGA profiles during thermal decomposition under N₂. **c** Tensile stress–strain curves of compression-molded *r*PO, commercial POE and OBC. **d** Lap-shear load- displacement curves measured by tensile testing. The inset illustrates the lap shear experiment setup. Pre-fix *c* denotes commercial polyolefins without the ester linkages.

*r*PO$_{3.1}$ (Supplementary Figs. 72 and 73). While *r*PO$_{12.2}$ presented broader orthorhombic reflections (110 and 200) with the weakening of amorphous phase reflection, *r*OBC$_{7.7}$ showed both sharp and diffused peaks corresponding to the crystalline and amorphous regions of the polymer, respectively (Supplementary Fig. 74). Furthermore, thermo-gravimetric analysis (TGA; Fig. 2b) of a few representative samples indicated desired thermal stability, with a high degradation temperature at 5% mass loss ($T_d$) above 400 °C, which is close to that of the commercial PE. Overall, the tunable thermal properties of these ester-linked ethylene copolymers were retained with installation of a few ester linkages along the polyolefin backbone.

**Mechanical Properties.** As shown in Fig. 2c, a series of ester-linked *r*PO and *r*OBC synthesized in this study retained advantageous mechanical properties analogous to commercial PE materials. Specifically, *r*PO$_{3.1}$ showed analogous tensile properties with yield stress ($\sigma_y$) of 9.3 MPa, an ultimate tensile strength at break ($\sigma_b$) of 15.6 MPa, and an elongation at break ($\varepsilon_b$) of 1021%, with Young's modulus ($E$ = 214.2 MPa) similar to commercial LLDPE ($E$ = 215.4 MPa; Supplementary Table 2). The *r*PO$_{8.9}$ and *r*PO$_{9.6}$ with a high branch density exhibited excellent soft elastomeric properties with $\sigma_b$ of 17.8 MPa and 14.0, high $\varepsilon_b$ of about 2233% and 2335%, and the tensile strength at 100% strain ($\sigma_{100\%}$) of 4.1 and 3.1 MPa, respectively. Notably, higher $\sigma_b$ of 24.9 MPa, $\sigma_{100\%}$ of 4.3 MPa, and $\varepsilon_b$ of 1953% were observed for the multiblock copolymer *r*OBC$_{7.7}$, suggesting that introducing hard (*t*PO$_{3.1}$) and soft (*r*PO$_{9.6}$) blocks via polycondensation assembly simultaneously enhanced hardness and tensile strength of the block copolymer relative to the random copolymer analog.

**Adhesion properties.** As installation of ester linkages to the polyolefin backbone to render chemical circularity should also bring about performance advantages in adhesion strength towards polar surfaces, we investigated adhesive properties of *r*PO and *r*OBC to polar surfaces. Polyether ether ketone (PEEK) and polyethylene terephthalate (PET) slides were chosen as representative polar surfaces. A series of single-lap joints between two PEEK slides were prepared using *r*PO$_0$−1, *r*PO$_{3.1}$, *r*PO$_{9.6}$, *r*OBC$_{7.7}$ and *c*PO$_{11.2}$ for lap shear experiments (Fig. 2d). *r*PO$_0$−1 and *r*PO$_{3.1}$ demonstrated strong adhesion to PEEK with an apparent lap shear force ($F_s$) of 244 N and 208 N, respectively. Likewise, *r*PO$_{8.9}$ and *r*PO$_{9.6}$ also showed strong adhesion to PEEK with $F_s$ = 210 and 201 N, respectively. In contrast, the adhesion of nonpolar commercial *c*OBC$_{10.4}$ to PEEK is much weaker ($F_s$ = 18 N; Supplementary Table 3). Notably, the block copolymer *r*OBC$_{7.7}$ exhibited excellent adhesion to PEEK surfaces with $F_s$ = 250 N. Changing the slide from PEEK to PET, *r*PO$_{9.6}$ also exhibited enhanced adhesion ($F_s$ = 208 N) to PET surface, which is more than 6 times higher than commercial *c*PO$_{11.2}$. These results indicate that the ester-linked olefin copolymers exhibited markedly enhanced adhesion to PEEK and PET surfaces relative to non-polar commercial polyolefins, reflecting the synergistic effects of the ester functionalities.

**Closed-loop recycling of ester-linked olefin copolymers**
The purpose of installing ester linkages to the polyolefin backbone is to render their closed-loop chemical recycling via transesterification. We first demonstrated this desired end-of-life feature by using *r*PO$_{9.6}$ as an example. Complete depolymerization was achieved by immersing *r*PO$_{9.6}$ in MeOH at 150 °C within 24 h under catalyst-free conditions. Upon cooling to room temperature, the recycled *t*PO$_{9.6}$ was recovered in essentially quantitative yield (> 98%) by simple filtration. Compared with the initial *t*PO$_{9.6}$ macromonomer, the $M_w$ and Đ of the recovered *t*PO$_{9.6}$ remained almost constant (Fig. 3a), and ¹H NMR analysis suggests the AB chain-end groups, -OH and -COCH₂CH₂CO₂R (R = Me or Et), remained intact after chemical recycling (Fig. 3b). Note that the R group was changed from ethyl (CH₃$CH_2$O-, $\delta$ = 3.68 ppm) to

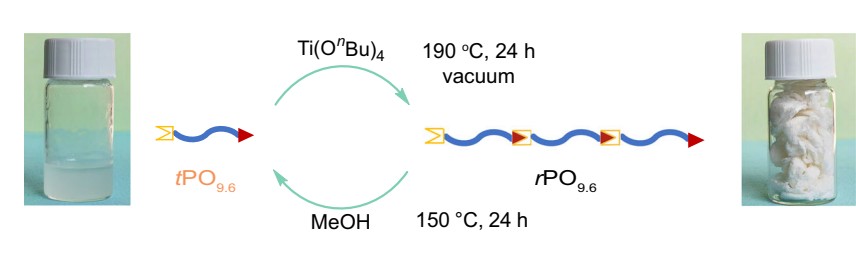

| Depolymerization cycle | $M_w$ | Đ | Repolymerization cycle | $M_w$ | Đ | $T_m$/°C |
|---|---|---|---|---|---|---|
| 0 | 6.3 | 1.3 | 0 | 90.6 | 2.3 | 77.0 |
| 1 | 7.3 | 1.4 | 1 | 100.9 | 2.7 | 76.8 |
| 2 | 7.2 | 1.5 | 2 | 103.8 | 2.8 | 76.9 |

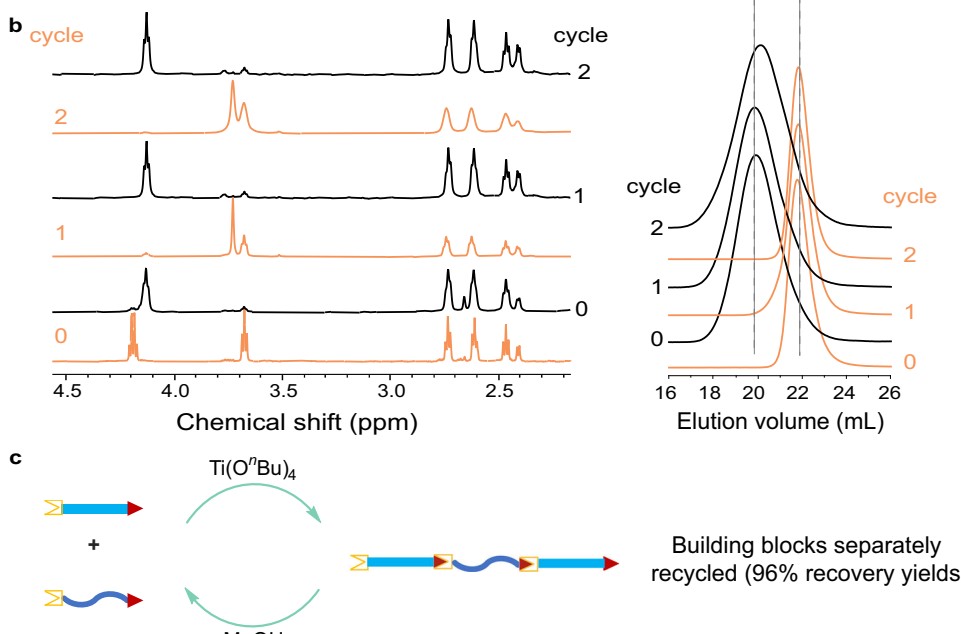

**Fig. 3 | Closed-loop chemical recycling demonstration. a** Depolymerization of $r\mathrm{PO}_{9.6}$ and repolymerization of the recovered $t\mathrm{PO}_{9.6}$ in multiple cycles. **b** $^{1}$H NMR and GPC traces of the original and recycled $r\mathrm{PO}_{9.6}$ (numbers and traces in black) and $t\mathrm{PO}_{9.6}$ (numbers and traces in orange). **c** Closed-loop recycling of $r\mathrm{OBC}_{9.4}$, demonstrating quantitative recovery of both soft and hard AB telechelic building blocks, $t\mathrm{PO}_{14.8}$ (96 % yield) and $t\mathrm{PO}_0$–1 (96% yield).

methyl (*Me*O-, $\delta = 3.73$ ppm) near quantitatively after methanolysis, and this change did not influence the closed-loop recycling process. With the recycled telechelic building block $t\mathrm{PO}_{9.6}$, repolymerization was conducted under our standard polycondensation conditions, and the resulting polymer had similar $M_w$ and Đ values as well as almost identical $^{1}$H NMR, compared with the initial polymer $r\mathrm{PO}_{9.6}$. Moreover, the repolymerized $r\mathrm{PO}_{9.6}$ showed a similar $T_m$ value to the initial polymer.

Next, we further investigated chemical recycling of $r\mathrm{OBC}$ as separately recovering the mixed telechelic building blocks is expected to be more complicated than the above single telechelic building block situation. Remarkably, the depolymerization process is still highly efficient, and recovering the two telechelic $t\mathrm{PO}$ building blocks was achieved via a simple separation procedure after the depolymerization reaction (Fig. 3c). Attributed to the dramatically different solubilities between linear $t\mathrm{PO}_0$–1 and branched $t\mathrm{PO}_{14.8}$, the former is barely soluble even in hot toluene, but the latter is well soluble even in *n*-

hexane. Extraction/filtration with *n*-hexane enables near quantitative recovery of pure $t\mathrm{PO}_{14.8}$ (96 % yield), and the remaining fraction is pure $t\mathrm{PO}_0$–1 (96% yield), as confirmed by NMR (Supplementary Figs. 26 and 27) and GPC (Supplementary Figs. 48 and 49).

Overall, we designed and synthesized ester-linked LLDPE-, POE-, and OBC-type copolymers by assembling a series of PE-based AB telechelic building blocks end-capped with -OH and -CO$_2$Et via tandem coordination and condensation polymerization methods, using directly abundant ethylene and 1-octene as feedstock monomers. Their high property-tunability and microstructural similarity to various commercial PE-based materials render them to possess similar thermal and mechanical properties, while the installed ester linkages de novo in the new $r\mathrm{PO}$ and $r\mathrm{OBC}$ materials bring about desired end-of-life chemical circularity and advantageous polar-surface adhesion properties. Meanwhile, it is worth noting that this CCTP-based method still needs to solve a series of underlying challenges in achieving near-quantitative efficiency chain initiation, chain transfer, quenching

processes, and the synthesis of various high difunctional purity telechelic building blocks without post-reaction separation/purificationn. Substantial efforts, especially in a catalytic perspective, are expected to significantly enhance the efficiency and make it a scalable and practical process in the future. Hence, this AB telechelic building-block strategy paves a new way for the design and synthesis of a wide range of circular olefin copolymers and hybrid polyolefin materials with closed-loop recyclability.

## Methods
### Materials
Toluene, *n*-hexane, diethyl ether (Et₂O) and tetrahydrofuran (THF) were distilled under N₂ and dried over Na/K alloy. Unless stated otherwise, all reagents were used as received. 6-Bromo-1-hexanol was purchased from Innochem (99%) and used after distillation. Ethyl 3-(chloroformyl)-propionate (97%) was purchased from Sigma-Aldrich and used after distillation. $Ti(O^nBu)_4$ (97%) and tetrabutylammonium fluoride (TBAF, 1.0 M in THF) were purchased from TCI, triisopropylsilyl chloride (TIPSCl) from Bide Chemical (98%), MeOH ($\geq$ 99.8%) from J&K Chemical, and toluene ($\geq$ 99.9%) from Shanghai Lingfeng, 1,1,2,2-Tetrachloroethane-$d_2$ (TCE-$d_2$) was purchased from Cambridge Isotope Laboratories and dried over molecular sieves. $C_6D_6$ (99+ atom % D) was purchased from Cambridge Isotope Laboratories and dried over Na/K alloy. Ethylene was purified by passage through an oxygen/moisture trap. Reagents (6-bromohexyl)oxytriisopropylsilane and $Zn[(CH_2)_6OTIPS]_2$ were synthesized according to literature procedures[42,43]. Ligand $^tBu$-ONPyrO and catalyst $Zr[^tBu$-ONPyrO]Bn₂ were synthesized according to literature procedures[44]. The co-catalyst $(C_{16}H_{33})_2NPhH^+B(C_6F_5)_4$ was obtained from Shanghai Chemspec Corp as a generous gift. High-density polyethylene (HDPE) (DOW DMDA-8904 NT 7), Linear Low-Density Polyethylene (DOW Tuflin™ HSE-1003 NT 7), Polyolefin elastomers (DOW Engage™ PV 8669, $cPO_{11.2}$) and Olefin Block Copolymer (DOW Infuse™ 9017, $cOBC_{10.4}$) were used as received.

### Instruments and characterizations
Unless stated otherwise, all manipulations were performed in an N₂-filled Vigor glove box or with standard Schlenk techniques. ¹H and ¹³C NMR spectra were recorded on Bruker Avance NEO 600 spectrometers. Data are presented in the following sequence: chemical shift, multiplicity, coupling constant in Hertz (Hz), and integration. Chemical shifts were referenced to the signal of the solvent. Molecular weights and dispersity (*Đ*) of the polymers were determined by high-temperature gel permeation chromatography (GPC) in 1,2-dichlorobenzene at 160 °C on a Polymer Char GPC-IR instrument, equipped with PSS Polefin Linear XL columns (3 × 30 cm, additional guard column), an infrared detector (IR5 MCT, concentration signal) and a viscosity detector. The flow rate was kept at 0.5 ml min⁻¹. Molecular weights were determined via universal calibration versus polystyrene standards. Differential scanning calorimetry (DSC) measurements of polymers were carried out on a TA DSC 2500 instrument with a heating/cooling rate of 10 °C min⁻¹. All the $T_m$ and $T_g$ values were obtained from the second heating scan. Decomposition temperatures ($T_d$, defined by the temperature at a 5% weight loss) of the polymers were measured using thermal gravimetric analysis (TGA) on a Q50 TGA Analyzer (TA Instrument). Polymer samples were heated from ambient temperature to 600 °C at a heating rate of 10 °C/min under N₂ flow. Powder X-ray diffraction (PXRD) studies were performed using a Rigaku Smartlab XRD spectrometer (9 KW) for Cu $K_\alpha$ radiation ($\lambda$ = 1.5406 Å), with a scan speed of 10° min⁻¹ and a step size of 0.02° in 2θ.

*Matrix-Assisted Laser Desorption/Ionization Time-of-Flight (MALDI-TOF) mass* characterization was conducted on a Bruker UltrafleXtreme TOF/TOF mass spectrometer (Bruker Daltonics, Inc., Billerica, MA) equipped with a Nd: YAG laser (355 nm). Dithranol (TCI, >95%) or trans-

2-[3-(4-tert-butylphenyl)−2-methyl-2-propenylidene] malononitrile (DCTB, TCI, >98%) was applied as the matrix. Sodium trifluoroacetate (CF₃COONa) was used as cationizing agent. The matrix in CHCl₃ at 20 mg/mL and the cationizing agent in ethanol at 20 mg/mL were mixed with the ratio of 10:1 (v/v). Each sample was prepared by depositing 0.5 μL of matrix solution on the wells of a 384-well ground-steel plate, allowing the spots to dry, depositing 0.5 μL of the sample on a spot of dry matrix, and adding another 0.5 μL of matrix on top of the dry sample. The plate was inserted into the MALDI source after drying. The mass scale was calibrated externally using polymethyl methacrylate at the molecular weight range under consideration in reflectron mode. Then, samples were tested in reflectron positive mode. And the data analysis was conducted with Bruker's FlexAnalysis software.

Tensile stress-strain test was performed on an Instron 5966 universal testing system (100 N load cell) using dog-bone-shaped samples (ASTM D638 standard, Type V) at a strain rate of 100 mm/min (10 mm/min for $rPO_0-1$). Polymer films with thickness of 0.6 – 1.0 mm were prepared via a compression molding process. Isolated polymer resins loaded between non-stick Teflon sheets were pre-compressed under 2.5 MPa and 150 °C for 8 min, compressed under 10 MPa and 150 °C for 2 min, and quenched under 2.5 MPa and 25 – 45 °C for 5 min. ASTM D638-5 standard dog-bone shaped samples were cut from the compression molded films. Thickness (0.8 ± 0.2 mm), width (2.0 ± 0.1 mm), and gauge length (10.0 ± 0.2 mm) of the measured dog-bone samples were used for normalization of the measured original data. Thickness (2.0 ± 0.2 mm), width (4.0 ± 0.1 mm), and gauge length (20.0 ± 0.5 mm) of the measured dog-bone samples were used for Young's modulus testing. Average values of 3–4 sample measurements with highly reproducibility were recorded in the tensile stress-strain curves (Fig. 2c), and the average values of 5–6 sample testing for tensile properties are listed in Supplementary Table 1.

For lap shear test, PEEK and PET sheets (100 × 10 × 0.5 mm) were initially washed with ethanol, and dried. Lap bonds were formed by placing a 10 × 10 × 0.6 mm film of $rPOs$, $rOBC$, $cPO$ or $cOBC$ sample between two PEEK (or PET) films and heating the sample to 150 °C under 2.5 MPa for 8 min and quenching under 2.5 MPa and 25 – 45 °C for 5 min. After cooling to room temperature, lap shear analysis was conducted using Instron 5966 universal testing system (1000 N load cell) with an extension speed of 5.0 mm/min at room temperature. Average values of 2−3 sample measurements with high reproducibility were used for lap-shear load-displacement curves in Fig. 2d. Average values of 3−4 sample testing for lap-shear tensile properties are summarized in Supplementary Table 3.

Calculation of octene content for the telechelic macromonomers before silyl deprotection. The results from high temperature ¹H NMR and ¹³C NMR are very similar, and we used ¹H NMR for all the calculations. Here is a representative example ($tPO_{8.9}$) to explain the calculation method: Octene content = $(T_A/3) \div [T_B-(T_A/3) \times 13]/4 + T_A/3) \times 100\% = 8.9$ mol% based on ¹H NMR (Supplementary Fig. 1). Octene content = $T_B \div [T_B + 0.5T_F-1.25T_E + 0.75T_G + T_H] \times 100\% = 9.0$ mol% based on ¹³C NMR (Supplementary Fig. 2)[45].

### Synthesis and characterization of *t*POs
**Ethylene homopolymerization.** In a typical experiment, a 150-mL oven dried glass pressure vessel equipped with a large stir bar was charged with dry toluene (100 mL), 20 μmol catalyst, 22 μmol co-catalyst $(C_{16}H_{33})_2NPhH^+B^-(C_6F_5)_4$ and $Zn[(CH_2)_6OTIPS]_2$ (1 mmol, 2 mL in *n*-hexane) inside a glovebox. The pressure vessel was sealed, removed from the glovebox, and attached to a high-pressure/high-vacuum line. The mixture was cooled to −78 °C in a dry ice/acetone bath, degassed, filled with ethylene gas, sealed, and then allowed to warm to the required temperature with an external bath. A solution of catalyst/cocatalyst was quickly injected into the flask with rapid stirring using a gas-tight syringe under N₂. The reactor was pressurized to

the required ethylene pressure. After the required reaction time, ethyl 3-(chloroformyl)propionate (0.988 g, 6.0 mmol) was injected into the pressure vessel and sealed for overnight reaction at 120 °C. The reactor was then vented and 500 mL MeOH was added to quench the reaction and precipitate the polymer. After stirring for 3 h, the polymer was collected by filtration, washed with MeOH, and dried under high vacuum at 60 °C overnight until a constant weight was obtained.

**Ethylene/1-octene copolymerization.** In a typical experiment, a 150-mL oven dried glass pressure vessel equipped with a large stir bar was charged with dry toluene (80 mL), 1-octene (60 mmol, 6.732 g) and $Zn[(CH_2)_6OTIPS]_2$ (50 equiv, 2 mL, 0.5 M in $n$-hexane) inside a glovebox. The pressure vessel was sealed, removed from the glovebox, and attached to a high-pressure/high-vacuum line. The mixture was cooled to −78 °C in a dry ice/acetone bath, degassed, filled with ethylene gas, sealed, and then allowed to warm to the required temperature with an external bath. A toluene solution of catalyst/cocatalyst was quickly injected into the flask with rapid stirring using a gas-tight syringe under $N_2$. The reactor was pressurized with 5 atm ethylene pressure. After the required reaction time, the ethyl 3-(chloroformyl)propionate (0.988 g, 6.0 mmol) was injected into the pressure vessel and sealed for the overnight reaction at 120 °C. The reactor was then vented and 500 mL MeOH was added to quench the reaction and precipitate the polymer. After stirring for 3 h, the polymer was collected by filtration, washed with MeOH, and dried under high vacuum at 60 °C overnight until a constant weight was obtained.

**Deprotection.** Purification of the telechelic macromonomers were conducted by flash chromatography prior to the deprotection step, which could greatly enhance the purity of the copolymer samples. For the telechelic macromonomers with no or low octene incorporation, toluene was used as the eluent; for the telechelic macromonomers with high octene incorporation, mixed hexane/toluene was used as the eluent. To a stirred solution of the purified telechelic macromonomers dissolved in toluene was added 10% (v/v) HCl/EtOH. The resulting mixture was stirred at 60 °C for 24 h. Solvent was removed under reduced pressure and the residue was washed with MeOH (250 mL × 3) to afford the hydroxy-terminated telechelic macromonomers.

**Synthesis of recyclable polymers $r$POs**
For the telechelic macromonomer polycondensation to afford $r$POs polyolefin, the telechelic macromonomer (1.0 equiv.) was dried in a two-necked Schlenk tube at 60 °C under vacuum. A toluene solution (0.03 M) of $Ti(O^nBu)_4$ (0.05 mol% vs. telechelic macromonomer) was added and the temperature was raised to 190 °C (stirring at 200 rpm). Oligomerization commenced, and vacuum was gradually applied (900 mbar to 2 mbar) over the course of 3 h. The polymerization step was conducted at 190 °C for typically 24 h. The resulting polymer was dissolved at 160 °C in xylene, precipitated in −30 °C with isopropyl alcohol, filtered, washed with MeOH, and dried under high vacuum at room temperature overnight until a constant weight was obtained, all the yields of the polymers were at least 98%.

**Depolymerization and repolymerization experiments**
**Depolymerization procedure.** For $r$PO$_{9.6}$. To a 150-mL oven-dried glass pressure vessel equipped with a large stir bar was added dry MeOH (60 mL) and the polymer $r$PO$_{9.6}$ (5.420 g). The depolymerization was carried out at 150 °C for 24 h. Upon cooling, the telechelic macromonomer precipitated, filtered, washed with MeOH, and dried under high vacuum at room temperature overnight until a constant weight was obtained (5.321 g, recovery yield > 98%).

For $r$OBC$_{9.4}$. To a 150-mL oven-dried glass pressure vessel equipped with a large stir bar was added dry MeOH (30 mL) and the polymer $r$OBC$_{9.4}$ (1.660 g). The depolymerization was carried out at 150 °C for 24 h. Upon cooling, the telechelic macromonomers precipitated, filtered, and washed with MeOH. Then add 50 mL 1-hexane, stir for 30 min at room temperature to fully dissolve the recovered $t$PO$_{14.8}$ and filtrate the remaining polymer and wash with 1-henxane (3 × 50 mL). The solid powder obtained by filtration is the recovered $t$PO$_0$−1 (0.565 g, recovery yield 96%). The filtrate was dried under reduced pressure to remove the solvent to obtain the recovered $t$PO$_{14.8}$ (1.055 g, recovery yield 96%).

**Repolymerization procedure.** The repolymerization step was the same as the standard ester polycondensation procedure (see "Synthesis of recyclable polymers $r$POs" section).

## Data availability
The data that support the finding of this study are present in the paper and/or the Supplementary Information and are available from the corresponding authors upon request.

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

## Acknowledgements

We are grateful for the financial support from the National Key R&D Program of China (2021YFA1501700; Y.G.), NSFC (U19B6001; Y.T.), Strategic Priority Research Program of the Chinese Academy of Sciences (XDB0610000; Y.G., Y.T.) and the start-up fund from Southern University of Science and Technology Guangdong Provincial Key Laboratory of Catalysis (2020B121201002; Y.T.). The work done at Colorado State University was supported by RePLACE (Redesigning Polymers to Leverage A Circular Economy) funded by the Office of Science of the U.S. Department of Energy (DE-SC0022290; E. Y.-X. C.). We would like to acknowledge the support from SUSTech CRF and the Analytical Instrumentation Center (SPST-AIC10112914), SPST, ShanghaiTech University. We thank Dr. Xiaopeng Li, Dr. Zhikai Li and Dr. Heng Wang from Shenzhen University for their great assistance in MALDI-TOF characterizations. We also thank Dr. Jing Tang for her helpful discussions and great assistance in preparing the figures. This work is dedicated to the 100th birthday of Professor Dr. Lixin Dai.

## Author contributions

Y.T., Y.G. and X.-L.S. conceived the project; Y.T and Y.G. directed research. Y.G., X.-W.H., Y.Y. Z. and X.-L.S. designed the experiments; X.-W.H. performed the experiments; X.Z. performed part of the experiments; A.M. (mechanical and adhesion) and P.L. (thermal) carried out the polymer property studies; B.Z. synthesized the Zr catalyst; X.K. checked all the data; Y.T. and E.Y.-X. C. provided insightful suggestions in the analysis and discussion of the results; X.-W.H., X.Z., and Y.G. wrote the initial manuscript draft. Y.T. and E.Y.-X. C. edited and revised the manuscript. The manuscript was approved by all the authors.

## Competing interests

X.-W.H., X.Z., Y.Y.Z., Y.G. and Y.T. are inventors of two provisional patent applications (CN 2023102055385 and CN 2023102055366, filing date: March 06, 2023). All other authors declare no competing interests.

## Additional information

**Peer review information** : *Nature Communications* thanks Jean Raynaud and the other, anonymous, reviewer(s) for their contribution to the peer review of this work. A peer review file is available.

