## [Peer Review File · Nature Communications]

Circular olefin copolymers made de novo from ethylene and α -olefinsEditorial Note: This manuscript has been previously reviewed at another journal that is not operating a transparent peer review scheme. This document only contains reviewer comments and rebuttal letters for versions considered at Nature Communications.

Reviewers' Comments:

Reviewer #1:

Remarks to the Author:

I commend the authors on their revised version of this manuscript. It is greatly strengthened for publication. However, though many of the data-based and experiment detail-based revisions were addressed adequately, the overall novelty and impact of this article remains below that of which I expect for publication in Nature Comm. For example, the authors have provided a response to my, and other reviewers, comments regarding this. However, their response can be distilled down to the fact that the chemistry described is not new, but that the authors simply found and used a slight variation to yield high purity species. I will reiterate that this chemistry is thorough and indeed interesting, and is certainly deserving of publication in a flagship journal (Macromolecules, Polymer Chemistry, ACS Catalysis, etc), but it simply does not meet the requirements that are required for publication in this, and other, highest tier magazines/journals

Reviewer #2:

Remarks to the Author:

Review of « Circular olefin copolymers made de novo from ethylene and α -olefins

The article entitled "circular olefin copolymers made de novo from ethylene and α -olefins" offers an interesting strategy to the issue of designing valuable circular polyolefins, or more likely circular additives for compounding (mainly due to economical reasons and associated costs of the overall proposed strategy). The problem it addresses (polyolefin or PO waste, and PO ecodesign) is certainly worthy of publication in general-audience and broad-readership journals, optionally or de facto open-access, such as Nature Communications.

The authors have greatly improved the quality of the manuscript since their initial submission.

Some important comments:

- If you use a "flash chromatography" on polymers, even long oligomers in that particular case, it is very unlikely it is a "straightforward" method. The experiment should be described to explicit how it can greatly increase the overall functionality of the telechelics. This is a clear limitation of the technique and claiming perfect telechelics (99% purity in the case of copolymers) without showing how you extensively purify them to achieve this incredible feat is insufficient. In addition, a comment should be made on how this is far from industrializable. Again, ^1H NMR (and maybe ^{13}C , which can be quite quantitative with the appropriate pulse sequence), before and after purification could enlighten the reader.
- If you describe MALDI-TOF spectrometry, you should highlight all populations (even the minor products), including the Me-terminated chains, which should also show up, unless their flight is hampered by less favorable functionality for ionization. In that later case, it should warrant a comment on how MALDI-TOF can induce some bias on evidencing chain-end fidelity.
- Looking at the SEC traces of the first entries, one can notice the shouldering, likely characteristic of different chain lengths, perhaps different termination reactions or at least imperfect polymerization processes. No real comment is provided there, other than the fact that in those cases, difunctional purity is lower.
- It is important to show the limits of a strategy in terms of circularity, and the authors now made an effort in both showing the pros and cons of such an approach.

After addressing these concerns, and thus minor revisions, I reckon this article could warrant publication in Nature Communications.

Reviewer #3:

Remarks to the Author:

I have read the revised version of the manuscript and the answers provided by the authors. I am satisfied with all of it, and I support publication in the journal Nature Communications.

A point-by-point response to all Editorial and Reviewer questions and comments

Reviewer #2 (Comments for the Author):

Question/comment 1. If you use a “flash chromatography” on polymers, even long oligomers in that particular case, it is very unlikely it is a “straightforward” method. The experiment should be described to explicit how it can greatly increase the overall functionality of the telechelics. This is a clear limitation of the technique and claiming perfect telechelics (99% purity in the case of copolymers) without showing how you extensively purify them to achieve this incredible feat is insufficient. In addition, a comment should be made on how this is far from industrializable. Again, ^1H NMR (and maybe ^{13}C , which can be quite quantitative with the appropriate pulse sequence), before and after purification could enlighten the reader.

Response: Thank you for the comments. The word “straightforward” was removed throughout the revised manuscript. A sentence “which could greatly enhance the purity of the copolymer samples.” was added as suggested in “Methods” section on Page 13 of the revised manuscript. Moreover, we already explicitly stated that “a flash chromatography was conducted for purification of the telechelic copolymers, which efficiently removed the impurities and achieved high difunctional purity” on Page 4 of the revised manuscript.

We totally agree with you regarding the prospect and future of using CCTP method for the synthesis of telechelic polyolefins. Besides the comments added in the conclusion section from the last round of revision, we added “Substantial efforts, especially in a catalytic perspective, are expected to significantly enhance the efficiency and make it a scalable and practical process in the future” in the conclusion section of the revised manuscript (Page 8).

Thank you for the suggestion on adding the NMR for a better comparison of the telechelic purity of the polymer before and after purification. ^1H NMR of $t\text{PO}_{12,2}$ before deprotection (without purification) was added in the revised Supplementary Information as Supplementary Figure 12. The ^1H NMR of $t\text{PO}_{12,2}$ after deprotection was already given as Supplementary Figure 11. It is now very clear to see the difference between them.

Supplementary Fig. 12. ^1H NMR of $t\text{PO}_{12.2}$ before deprotection (without purification) in $\text{TCE-}d_2$ (110 $^\circ\text{C}$), Table 1 entry 7.

Question/comment 2. If you describe MALDI-TOF spectrometry, you should highlight all populations (even the minor products), including the Me-terminated chains, which should also show up, unless their flight is hampered by less favorable functionality for ionization. In that later case, it should warrant a comment on how MALDI-TOF can induce some bias on evidencing chain-end fidelity.

Response: Thank you for pointing this out. We did observe methyl chain end for telechelic samples with mono-telechelic impurity, and calculated the difunctional purity based on the integrations of the chain end groups, including methyl. While MALDI-TOF is good at characterizing telechelic polyolefin macromonomers, its effectiveness depends on the macromonomers' functionality as well as the conditions. We carefully examined the MALDI-TOF results but failed to identify such impure minor products. It might be due to the reason you mentioned and the low content of such impure polymer. The samples we used for the analysis were after purification by flash chromatography and thus the content of such impure polymer, if any, is very low.

In response, we added a sentence "We failed to observe the methyl-terminated macromonomer as the minor impurity, which likely reflects the MALDI-TOF bias on evidencing chain-end fidelity especially for the samples with high purity" on Page 4 of the revised manuscript.

Question/comment 3. Looking at the SEC traces of the first entries, one can notice the shouldering, likely characteristic of different chain lengths, perhaps different termination reactions or at least imperfect polymerization processes. No real comment is provided there, other than the fact that in those

cases, difunctional purity is lower.

Response: We thank you for the insightful comment! It is indeed an important detailed information that we should explore in depth in follow-up research. We carefully examined the characterizations of the telechelic macromonomer *t*PO₀-1, and the GPC trace is a monomodal polymer with narrow dispersity (see the figure below). However, after TIPS deprotection, the small shoulder peak showed up in the GPC trace (Supplementary Fig. 28 in the revised Supplementary Information on Page S17). As the macromonomer now bears an -OH at the chain end, which could possibly undergo esterification reaction during the test, as the polymer needs to be dissolved in *o*-dichlorobenzene at 160 °C for at least 2 hours. The transesterification process was confirmed by parallel ¹H NMR characterizations.

In response, we added the sentence “The shoulder peak likely arises from the esterification of the chain-end -OH and -CO₂R groups during the sample preparation and measurement” in the figure caption of Supplementary Fig. 28 (Page S17 of the revised Supplementary Information).

SEC trace of *t*PO₀-1 before deprotection, Table 1 entry 1.

Question/comment 4. It is important to show the limits of a strategy in terms of circularity, and the authors now made an effort in both showing the pros and cons of such an approach.

Response: We thank you very much for the great suggestions and comments from last and this rounds, which significantly enhanced the manuscript. It is now a more comprehensive piece of work, which outlined the whole picture of this strategy.